# The Efficiency of Serum Biomarkers in Predicting the Clinical Outcome of Patients with Mesenteric Ischemia during Follow-Up: A Systematic Review

**DOI:** 10.3390/diagnostics14070670

**Published:** 2024-03-22

**Authors:** Florin Vasile Mihaileanu, Stefan Lucian Popa, Simona Grad, Dinu Iuliu Dumitrascu, Abdulrahman Ismaiel, Eliza Rus, Vlad Dumitru Brata, Alexandru Marius Padureanu, Miruna Oana Dita, Daria Claudia Turtoi, Traian Adrian Duse, Andrei Vlad Badulescu, Paolo Bottalico, Giuseppe Chiarioni, Cristina Pop, Cristina Mogosan, Maria Barsan, Claudia Diana Gherman, Bogdan Stancu, Liliana David

**Affiliations:** 1Department of Surgery, County Hospital, “Iuliu Hatieganu” University of Medicine and Pharmacy, 400139 Cluj-Napoca, Romania; ms26rfl@yahoo.com (F.V.M.); bstancu7@yahoo.com (B.S.); 22nd Medical Department, “Iuliu Hatieganu” University of Medicine and Pharmacy, 400000 Cluj-Napoca, Romania; costinsimona_m@yahoo.com (S.G.); abdulrahman.ismaiel@yahoo.com (A.I.); lilidavid2007@yahoo.com (L.D.); 3Department of Anatomy, “Iuliu Hatieganu” University of Medicine and Pharmacy, 400000 Cluj-Napoca, Romania; d.dumitrascu@yahoo.com; 4Faculty of Medicine, ”Iuliu Hatieganu” University of Medicine and Pharmacy, 400000 Cluj-Napoca, Romania; bordea.eliza@gmail.com (E.R.); brata_vlad@yahoo.com (V.D.B.); alexandru.padureanu@outlook.com (A.M.P.); miruna.dita@outlook.com (M.O.D.); turtoidariaclaudia@gmail.com (D.C.T.); adrianduse@yahoo.com (T.A.D.); andibadulescu@gmail.com (A.V.B.); 5Department of Medical Sciences, University of Turin, 10126 Turin, Italy; paolo.bottalico@edu.unito.it; 6Division of Gastroenterology B, AOUI Verona, University of Verona, 37126 Verona, Italy; chiarioni@alice.it; 7Department of Pharmacology, Physiology, and Pathophysiology, Faculty of Pharmacy, Iuliu Hatieganu University of Medicine and Pharmacy, 400349 Cluj-Napoca, Romania; cristina.pop.farmacologie@gmail.com (C.P.); cmogosan@umfcluj.ro (C.M.); 8Department of Occupational Health, “Iuliu Hatieganu” University of Medicine and Pharmacy, 400000 Cluj-Napoca, Romania; maria.opritoiu@umfcluj.ro; 9Department of Surgery-Practical Abilities, “Iuliu Hațieganu” University of Medicine and Pharmacy Cluj-Napoca, 400337 Cluj-Napoca, Romania; gherman.claudia@umfcluj.ro

**Keywords:** mesenteric ischemia, biomarkers, clinical outcomes, follow-up

## Abstract

The initial clinical manifestation of acute mesenteric ischemia poses a diagnostic challenge, often leading to delays in identification and subsequent surgical intervention, contributing to adverse outcomes. Serum biomarkers, offering insights into the underlying pathophysiology, hold promise as prognostic indicators for acute mesenteric ischemia. This systematic review comprehensively explores the role of blood biomarkers in predicting clinical outcomes during follow-up for patients with mesenteric ischemia. A thorough literature search across the PubMed, Cochrane Library, and EMBASE databases yielded 33 relevant publications investigating the efficacy of serum biomarkers in predicting outcomes for mesenteric ischemia. Numerous studies underscore the utility of blood biomarkers in swiftly and accurately differentiating between causes of mesenteric ischemia, facilitating a prompt diagnosis. Elevated levels of specific biomarkers, particularly D-dimers, consistently correlate with heightened mortality risk and poorer clinical outcomes. While certain serum indicators exhibit substantial potential in associating with mesenteric ischemia, further research through rigorous human trials is imperative to enhance their consistent predictive ability during the follow-up period. This study underscores the diagnostic and prognostic significance of specific biomarkers for mesenteric ischemia, emphasizing the necessity for standardized procedures in future investigations.

## 1. Introduction

Acute mesenteric ischemia (AMI) is a syndrome caused by a sudden onset of mesenteric vessels hypoperfusion, secondary to a reduction in arterial or venous inflow, resulting in ischemia and gangrene of the bowel wall [1]. The pathogenesis includes arterial embolism, arterial thrombosis, venous thrombosis, and non-occlusive mesenteric ischemia (NOMI) [1,2,3].

AMI is subdivided into non-occlusive mesenteric ischemia (NOMI) and occlusive mesenteric arterial ischemia (OMAI), which is further subdivided into acute mesenteric arterial embolism (AMAE), and acute mesenteric arterial thrombosis (AMAT). Also, the venous disease takes the form of mesenteric venous thrombosis (MVT) [1,2,3]. Other causes include internal hernia with strangulation, volvulus, intussusception, tumor compression, aortic dissection and post-angiography thrombosis, and blunt trauma [1,2,3].

However, the clinical presentation with the classic triad of abdominal pain, fever, and heme-positive stools is present only in 33% of the cases [1]. The rest of the cases present atypical symptomatology, which includes nausea, emesis, anorexia, and diarrhea, progressing to constipation, abdominal distention, tachycardia, tachypnea, hypotension, fever, and finally, altered mental status. Because AMI is a syndrome with an unclear initial presentation, the differential diagnosis is difficult, and a significant number of both organic and functional digestive disorders have common symptoms. Consequently, the diagnosis and surgical treatments are considerably delayed, with fatal outcomes [2,3,4].

An early recognition of clinical particularities, followed by an accurate biological and imagistic diagnosis can reduce the high mortality risk. Current guidelines recommend CT angiography to diagnose AMI [1], but early CT findings are non-specific, especially in NOMI. The limited availability of CT angiography and the time-consuming management pose challenges, especially considering time as the primary prognostic marker in AMI [1,2,3].

In the realm of mesenteric ischemia, understanding the intricate biochemical pathways is crucial for unraveling the disease’s complexities. The biochemical processes underpinning mesenteric ischemia involve cascades of inflammation, oxidative stress, and cellular injury. Serum biomarkers, representing molecular fingerprints of these processes, offer a unique opportunity for prognostic insights.

The clinical outcomes of patients with mesenteric ischemia are highly variable, ranging from reversible symptoms to severe complications. As a result, there is a growing interest in identifying reliable biomarkers that can aid in predicting the clinical course and outcomes of individuals with mesenteric ischemia during follow-up.

Thus, because AMI is an acute disorder for which accurate diagnosis and treatment are needed, a reliable panel of plasma biomarkers could help avoid treatment delays due to atypical symptomatology, difficult differential diagnosis, and unpredictable clinical evolution. Many papers have been published evaluating the usefulness of plasma biomarkers in AMI, but clinicians also need studies that compare findings and studies that give them a broader view regarding the topic of plasma biomarker usefulness in AMI.

Serum biomarkers, with their potential to reflect underlying pathophysiological processes, have emerged as promising candidates for diagnostic and prognostic assessment.

For this reason, the objectives of this systematic review are to critically evaluate and synthesize existing evidence on the efficiency of serum biomarkers in predicting the clinical outcomes of patients with mesenteric ischemia during follow-up. To understand the relation between clinical outcome, pathogenesis, and differences between different biomarkers, we also analyzed the dynamics of the available biomarkers, including positivation time, sensitivity, specificity, and false positive results. By comprehensively assessing the current literature, this review seeks to contribute valuable insights that may inform clinical practice and guide future research efforts in optimizing the management of mesenteric ischemia.

## 2. Materials and Methods

This systematic review was written following the preferred reporting items for systematic review (PRISMA) guidelines [5].

### 2.1. Data Sources and Search Strategy

The electronic databases PubMed, EMBASE, and the Cochrane Library were searched without any restrictions, from their inception until 4 December 2023, to identify potential observational studies. The following search string was entered for PubMed (“ Mesenteric Ischemia” [Mesh]) OR (“Mesenteric Infarction” [All Fields]) AND (“Biomarkers” [Mesh]) OR (“D-dimer” [All Fields]) OR (“Endothelin-1” [All Fields]) OR (“I-FABP” [All Fields]) OR (“alpha-GST” [Mesh]) OR (“IL-6” [All Fields]) OR (“L-Lactate” [All Fields]) OR (“D-Lactate” [All Fields]) OR (“Procalcitonin” [All Fields]) OR (“Citrulline” [All Fields]) OR (“Ischemia Modified Albumin” [All Fields]) OR (“Omentin” [All Fields]) OR (“Prognosis” [All Fields]) OR (“Follow-Up” [All Fields]) OR (“Prediction” [All Fields]) OR (“Biomarkers” [All Fields]) OR (“Outcome” [All Fields]), and similar search terms were used for EMBASE and the Cochrane Library. Furthermore, in order to reduce the risk of bias, we manually sorted the pertinent results across the three databases, as shown in Figure 1.

### 2.2. Study Selection and Eligibility Criteria

Observational and interventional studies analyzing the impact of serum biomarkers in predicting the outcome of patients with mesenteric ischemia were included in the study.

Studies were included if (1) a potentially relevant biomarker was measured and assessed in patients with suspicion of mesenteric ischemia, (2) the diagnosis of mesenteric ischemia was confirmed in patients included in the studies, and if (3) the study had proper statistical methods of assessing the dynamic of the investigated biomarkers.

Studies were excluded if they were published in other languages than English, if they were case reports, letters to the editor, reviews, practice guidelines, conference abstracts/papers, abstracts published without the full text, or if the paper was unavailable. In addition, studies on pediatric populations were also excluded.

Two investigators (S.L.P. and A.I.) independently evaluated the titles, abstracts, and full texts of the papers resulting from the search. In case of discrepancies between the two investigators, a consensus was reached through discussion

To address the risk of bias of missing results, the two investigators carefully examined the included articles in terms of methodology and results to ensure consistency between planned and reported outcomes. Discrepancies between pre-specified and reported outcomes may indicate selective reporting bias, necessitating the cautious interpretation of the study findings.

One common form of missing results is publication bias, where studies with positive or significant results are more likely to be published than those with null or negative findings. This bias can skew the overall evidence base, leading to inflated effect sizes and misleading conclusions.

In parallel to assessing the risk of bias associated with missing results, determining the certainty of evidence is paramount in evaluating the strength and reliability of research findings. The two investigators mostly included randomized controlled trials (RCTs) in the analysis, with a low risk of bias and consistent, precise findings and excluded irrelevant articles.

### 2.3. Data Extraction

The following information was extracted from the included studies: names of the authors, publication year, country, number of subjects and study population, age, gender distribution, biomarker levels, mesenteric ischemia diagnosis, results, and main findings. The data were extracted and included into a spreadsheet by two investigators (F.M. and S.G.) and any discrepancies regarding the outcome of data extraction were resolved through discussion. The final data were aggregated into the current manuscript.

### 2.4. Quality Assessment

Conducting a quality assessment in a systematic review is a critical step to ensure the reliability and validity of the included studies. The quality assessment helps researchers evaluate the methodological rigor, risk of bias, and overall trustworthiness of individual studies, thereby informing the synthesis and interpretation of findings.

In order to perform the quality assessment, we clearly defined the criteria against which the quality of the included studies was assessed. These criteria typically related to study design, methodology, conduct, and reporting standards. Next, we developed a data extraction form that included fields for recording relevant information from each study, such as study design, participant characteristics, interventions, outcomes, and quality assessment items. The data extraction form must be comprehensive and structured to facilitate consistent and thorough extraction of information. Third, we trained our reviewers to ensure inter-rater reliability. Reviewers were familiarized with the quality assessment criteria and data extraction form, and discrepancies in assessments were resolved through discussion or consensus.

We conducted the quality assessment independently for each included study according to the predefined criteria. We evaluated various domains of study quality, such as selection bias and reporting bias. The results of the quality assessment for each study were recorded in the data extraction form. The findings of the quality assessment, across all included studies, were considered. We considered the overall risk of bias or the quality of evidence within the body of literature, taking into account the strengths and limitations of individual studies.

By following these steps, we were able to systematically assess the quality of the included studies in a systematic review, thereby enhancing the rigor and credibility of the review findings. The quality assessment plays a crucial role in synthesizing evidence, informing recommendations, and guiding evidence-based practice and policy decisions.

## 3. Results

As seen in Figure 1, a total of 341 records were identified, from which 184 were removed in the identification step. Out of the 157 papers screened, 111 were excluded based on the listed exclusion criteria. A total of 13 papers could not be retrieved and 33 papers were included in the review.

### 3.1. Biomarkers with Clinical Use


**Endothelin-1**


Endothelin-1 (ET-1) is a peptide produced by the endothelial cells in the vessel wall in response to different stimuli, such as cytokines, thrombin, or even sheer stress. High ET-1 levels have been identified as a risk factor for NOMI [6]. The duration required for this serum marker to become positive after NOMI is approximately 30 min. Groesdonk et al. conducted an observational cohort study, monitoring ET-1 levels before and after cardiac surgery [6]. The study concluded that elevated ET-1 levels both preoperatively and postoperatively predisposed patients to develop NOMI, the risk increasing with every picogram/milliliter [6]. Additionally, the team showed that monitoring ET-1 levels has a high accuracy in predicting NOMI [6]. Table 1 presents the results of this study in terms of the specificity and sensitivity of ET-1 as a biomarker for NOMI.


**I-FABP**


Intestinal fatty acid-binding proteins (I-FABPs) are cytosolic proteins that are present in the epithelial cells of the small intestine and in smaller amounts in the stomach, colon, and other organs, being released in the circulation when intestinal epithelial cells are damaged, and ultimately, they are excreted in the urine [7]. After ischemia is produced, this serum marker takes 60 min to become positive [8].

Studies on animal models showed that, given its structure and low molecular weight, I-FABP is released in the blood 15–30 min after ischemia onset, making this protein a promising marker for acute intestinal ischemia [9]. Urinary FABP levels might be even more promising because the clearance of plasma FABPs is very rapid, with an accumulation of FABPs in the urine and better accuracy for urinary tests [10].

In their study, Schellekens et al. showed that the duration of ischemia determined the tissue damage and correlated it with the level of I-FABPs [8]. The authors used an experimental model of jejunum exposed to 15, 30, or 60 min of ischemia, followed by reperfusion. The serum concentration of I-FABPs was 463.3 ± 139.8 ng/mL after 60 min of ischemia, compared to only 7.79 ± 1.8 ng/mL after 15 min of ischemia [8].

In a study with 61 critically ill patients with NOMI suspicion, Bourcier et al. discovered that median plasma I-FABP concentration was significantly higher in patients with definite intestinal necrosis than those with ruled-out diagnoses. ICU survival was strongly associated with intestinal resection, while no patient survived without necrotic resection. I-FABPs were significantly higher in patients with intestinal necrosis and in NOMI non-survivors, respectively [11]. In addition, Sekino et al. identified that I-FABPs at the level of ICU admission can serve as a predictor of 28-day mortality in septic shock patients and are associated with the incidence of NOMI and in-hospital deaths caused by abdominal complications [12].

Matsumoto et al. conducted a study analyzing the usefulness of monitoring I-FABPs, D-dimers, CRP, and lactate in patients with clinical suspicion of NOMI [13]. The study concluded that the patients diagnosed with NOMI exhibited elevated levels of the mentioned markers [13]. Although the best results were obtained by combining the monitoring of these markers, the best performance was achieved when the model included I-FABPs [13]. Guzel et al. assessed three biological parameters: leucocyte count, D-dimer levels, and serum I-FABP levels [14]. Only I-FABP levels were significantly higher in AMI patients compared to patients with other causes of acute abdomen [14]. Similar results were reported by Thuijls et al., who measured not only plasma but also the urinary concentrations of the three FABP isoforms [10]. All these proteins were significantly higher in patients with AMI than in patients with other diseases. Urinary FABP levels also increased positive post-test probabilities > 80% [10]. Kanda et al. evaluated 361 patients who underwent surgery for acute abdomen [15]. Using a cut-off value of 3.1 ng/mL for I-FABP, 78.8% of patients with AMI had serum I-FABP levels higher than the cutoff, compared to 26.3% among ischemia-free patients. I-FABP levels had the best AUC but lower specificity than serum LDH and CPK. Most patients without ischemia and increased levels of I-FABPs had small bowel disease [15]. This study confirmed that small bowel diseases determine an increase in serum I-FABP levels, thus, while serum I-FABP might not be very specific for bowel ischemia, it is specific for bowel disease. A significant drawback reported by the authors is that the test results were available within 3 h. Therefore, it would be inappropriate in an acute setting [15]. Table 1 presents the detailed results of these studies.


**Serum lactate**


Serum lactate is a valuable biomarker for predicting bowel necrosis length and mortality. Conde Monroy et al. conducted a retrospective cross-sectional study of patients with AMI undergoing laparotomy. They found a correlation between serum lactate admission levels (SLALs) with bowel necrosis length and mortality in patients with AMI. There was a significant association between SLAL and mortality [16]. Survival in patients with SLALs < 3.8 mmol/L could be determined by the bowel necrosis length, while for cases with SLALs > 3.8 mmol/L, survival could be determined by a bowel resection performed within 3.5 h after diagnosis. In patients with AMI, SLALs can be used as a prognostic tool for mortality prediction. Mortality was determined using bowel necrosis length, intestinal resection, and the time in which the surgical procedure was performed. Cases with necrosis lengths less than 177 cm had a 13% mortality, while those with longer necrosis had a mortality of 80% [16]. Table 1 presents the detailed result of this study. A retrospective study by Studer et al. with 91 patients affected by AMI reached the same conclusion: the highest serum lactate value measured 24 h before surgery showed a moderate correlation but no statistical significance with the length of bowel necrosis [17]. When compared to values > 6 h before surgery, there was a significant increase in the mean serum lactate levels and a decrease in the mean serum pH in the period 0 to 6 h before surgery. This finding was emphasized in patients with >50 cm of necrotic bowel. Furthermore, within 24 h before surgery, the non-survivors had statistically higher mean serum lactate values than the survivors. Forward logistic regression showed that the length of the necrotic bowel and the highest lactate value within 24 h before surgery were independent risk factors for mortality [17].

In a study by Martin et al., 119 patients with AMI were included and divided into two groups according to the time of diagnosis between the arrival at the emergency unit and the CT scan: early diagnosis and delayed diagnosis [18]. Patients with delayed diagnosis tended to be associated with lower rates of revascularization (9 vs. 17%), higher rates of major surgical morbidity (90 vs. 57%), a longer length of stay (16 ± 23 vs. 13 ± 15 days), and, at the end of follow-up, a higher rate of short, small bowel syndrome (18 vs. 7%). AMI was diagnosed with a 3-phase CT scan, allowing the identification of the mechanism and the severity. The arterial mechanism was significantly less frequently observed in the early diagnosis group than in the delayed diagnosis group. On the contrary, venous thrombosis was more frequently observed in the earlier than delayed diagnosis group, without significant difference. None of the findings about severity were statistically significant [18]. In addition, Destek et al. showed that there was no statistically significant difference between the groups concerning L-lactate and the producing enzyme, lactate dehydrogenase (*p* > 0.05) [19].

Murray et al. found that in AMI, the degree of intestinal wall injury notably increases the levels of D-lactate, and to a lesser extent, L-lactate [20]. The period needed for both stereoisomers to turn positive is around 5 min [21]. However, its reliability is impaired because serum D-lactate is linked to non-acute surgical diseases and dietary factors such as jejuno-ileal bypasses, short bowel syndromes, probiotic administration, and increased carbohydrate intake [22,23,24,25,26]. The lactate levels measured by current gasometry equipment are not differentiated between D- and L-lactate. Thus, it is recommended that future research concentrate on the distinct functions of D- and L-lactate as indicators of the severity of AMI and push for creating gasometry instruments that offer values specifically for each stereoisomer [27].


**Fibrinogen-to-Albumin Ratio**


The fibrinogen-to-albumin ratio (FAR) has been identified in several studies as a predictor of poor outcomes and adverse events in patients with cardiovascular diseases, sepsis, stroke, and cancer. Muhtaroğlu et al. investigated the prognostic value of FAR for the first time in patients with AMI and concluded that it may be a valuable prognostic biomarker. In the logistic regression analysis, the postoperative FAR value affected survival, while preoperative FAR did not [28]. The mean pre- and postoperative values of other variables were also examined. The mean pre- and postoperative fibrinogen levels were statistically significantly higher in the non-survivor group than in the survivor group (*p* < 0.001). As a result, the survivor group’s preoperative hemoglobin value was significantly higher than the non-survivor group (*p* = 0.059, *p* < 0.001; respectively). The non-survivor group’s preoperative WBC and neutrophil count were statistically significantly higher. Additionally, the non-survivor group had statistically higher postoperative CRP and D-dimer levels [28].


**C-reactive protein**


CRP has been proposed as a predictor of AMI diagnosis and prognosis. According to Muhtaroğlu et al., postoperative CRP levels were statistically significantly higher in the non-survivor group (*p* < 0.05). Preoperative CRP levels were not statistically significantly higher in the non-survivor group [28]. On the other hand, in a previous study by Destek et al., CRP levels were used efficiently in the operative period to diagnose AMI and characterize its subtype and clinical course. This study compared AMI types with auxiliary biochemical tests performed before and after surgery. The EAMI group had the highest CRP level, whereas the NOMI group had the lowest. The total length of stay in the hospital was found to be significantly correlated with CRP (*p* = 0.045) levels. In the analyses, CRP was determined to be the common biomarker that could be used in the diagnosis of mesenteric ischemia in all AMI types [19].


**Neutrophil-to-Lymphocyte ratio and Platelet-to-Lymphocyte ratio**


The neutrophil-to-lymphocyte ratio (NLR) prognostic value was investigated by Augène et al. for 30-day outcomes in 106 patients with AMI. To assess the potential interest of the PLR in patients diagnosed with AMI, the population was divided into four subgroups based on the PLR quartile [29]. The outcomes of patients were compared based on their PLR value at the time of in-hospital admission. The 30-day mortality rate in the IV group (PLR *>* 429.3) was significantly higher than in other groups. The NLR’s potential as a predictive factor was then assessed, and patients were classified into four other subgroups based on their NLR value. The outcomes of patients did not significantly differ among the subgroups. Even though patients in the IV group had a higher 30-day death rate than the other groups, the difference was not statistically significant [29]. The same conclusion about NLR was drawn by Destek et al.: there was no statistically significant difference between the groups concerning leukocytes and NLR values (*p* > 0.05) [19].


**Procalcitonin**


Cosse et al. demonstrated that procalcitonin (PCT) might be used as a necrosis marker, particularly in cases of extended damage, and that it reflects the patient’s prognosis. They established a gray zone to assess the conditions in which PCT would be insufficiently informative using the R software with the ROCR package [30]. Cosse et al. investigated the relationship between PCT levels and failure of the conservative management of patients with small bowel obstruction, concluding the fact that PCT was significantly higher in patients requiring surgical treatment and patients with intraoperative ischemia [31]. Table 1 presents the detailed results of this study. Additionally, in a further study, the team evaluated the PCT levels at admission and 18 and 24 h after admission to the hospital in patients benefiting from conservative or surgical treatment for small bowel obstruction, with the aim of better assessing the prognosis and evolution of conservative management and identifying patients that would benefit from more accurate and timely surgical treatment [32].

A 2011 study by Karabulut et al. [33] on three groups of seven New Zealand rabbits demonstrated a significant rise in PCT levels as soon as 1 h after SMA ligature and up to 6 h, demonstrating both early and late biomarker potential. A similar rat model from 2015 by Karaca et al. [34] partially challenged the previous preclinical results, showing a significant increase only in the later stages, after ischemia was induced (i.e., 6 h after debut). Nonetheless, preclinical studies demonstrated the clear use of PCT as a biomarker for intestinal ischemia.

The value of PCT in differentiating patients with and without intestinal ischemia and necrosis was investigated in several studies with significant results [30,31,35,36]. Nagata et al. used PCT levels as a negative screening test for colonic ischemia following open abdominal aortic surgery [35]. Additionally, another study assessed the value of PCT in detecting bowel ischemia and necrosis in patients presenting with bowel obstruction, concluding that PCT levels were independent predictors of ischemia and necrosis [36]. Table 1 presents the detailed results of these studies.


**Citrulline**


Cakmaz et al. conducted a preclinical study on 21 Wistar rats divided into three groups (control, short-term ischemia, and long-term ischemia), measuring the plasma levels of diamine oxidase (DAO) and citrulline [37]. The results showed a significant increase in DAO and a decrease in citrulline in both the short- and long-term ischemia groups compared to control, and a larger decrease in citrulline levels and a larger increase in DAO levels in the long-term ischemia group. A more recent study was conducted by Kulu et al. on 48 patients with acute abdominal complaints divided into two groups: with AMI and without AMI [38]. The findings demonstrated significantly lower citrulline levels in patients with AMI (n = 23) than those with other acute conditions. Moreover, the AUROC was the second highest for citrulline, surpassed only by that for D-dimers (0.72 and 0.85, respectively). Table 1 presents the detailed results of these two studies. Although encouraging, these results are contradicted by a 2021 cross-sectional study by Nuzzo et al. conducted on 129 French patients (50 with AMI and 79 controls) [39]. Unfortunately, the low sensitivity in both studies demonstrates poor diagnostic performance for citrulline in detecting AMI, considering the severe consequences in misdiagnosis cases.


**Interleukin 6**


In terms of using IL-6 as a reliable biomarker, studies show that this cytokine turns positive in one hour after intestinal ischemia [40]. In a prospective study, 56 patients were included in five groups: the bowel ischemia, bowel obstruction, acute inflammation, perforation, colorectal adenocarcinoma, and control group. In the multivariate analysis, IL-6 was an independent predictor of the definite diagnosis and differentiated bowel ischemia from the rest of the pathologies. The 27.66 pg/mL cut-off value for IL-6 had a 100% sensitivity and specificity [41]. Salim et al. reported significantly higher serum IL-6 levels in AMI patients compared to nonischemic control patients. Urine I-FABP was also significantly increased in AMI patients compared to controls [42].

Also, in an experimental study, New Zealand rabbits were randomly divided into three groups (controls, sham laparotomy, and superior mesenteric artery ligation), and it was revealed that the serum ischemia-modified albumin (IMA) and serum IL-6 levels in the ischemia group were significantly higher than those of the control and sham groups [40].


**Ischemia-modified albumin**


The potential utility of ischemia-modified albumin (IMA) in AMI diagnosis was first revealed in a preliminary case-control study by Gunduz et al. [43]. The results showed that the patients in the SMA occlusion group had significantly higher levels of IMA. Polk et al. conducted a more extensive study the same year, supporting the findings. Patients undergoing laparotomy for bowel obstruction diagnosed postoperatively with intestinal ischemia showed significantly higher levels of IMA compared to those without intestinal ischemia [44]. Detailed results of this study are presented in Table 1.

Further animal models were performed in later years to investigate whether the kinetics of IMA deemed it suitable as a marker in acute mesenteric ischemic events. Gunduz et al. demonstrated the kinetics of IMA on a Wistar rat model in 2009 [45]. IMA levels were significantly higher in the ischemia group than in the control early on after SMA ligature (30 min) and rose in parallel with the duration of ischemia. Therefore, the data suggest that IMA could be of potential use in the early detection of AMI. Dundar et al. reported findings that challenged the previous preclinical results in a study on 21 New Zealand rabbits [40]. The authors found that IMA levels were significantly higher in the ischemia group only after 3 and 6 h of SMA occlusion. Moreover, another animal model conducted by Uygun et al. on four groups of eight Wistar rats demonstrated that, although histological scores proved to be significantly different in the ischemia groups, IMA levels showed no significant increase, questioning the clinical use of IMA in both early and later stages of AMI [46].


**D-dimer**


Acosta et al. conducted a preliminary study to assess the relationship between D-dimer levels and acute bowel ischemia by monitoring the markers’ levels in patients with clinical suspicion of the disease [47]. This serum marker turns positive in 30 min. The study concluded that, for thromboembolism of the SMA, bowel ischemia of any other cause, and patients needing a laparotomy, the sensitivity and predictive value of a negative D-dimer test were 100% [48]. Additionally, in another study, with median levels of 1.6 mg/L, patients with AMI had significantly higher levels of D-dimer than patients in the control group [47].

D-dimer levels have also been analyzed in correlation with the severity of bowel necrosis resulting from AMI [49]. A study by Chiu et al. examined the diagnostic value of D-dimer levels in 67 patients, out of which 23 were further diagnosed with AMI [49]. The team concluded that detecting D-dimer in the serum did not significantly help differentiate patients with and without AMI [49]. Moreover, the study revealed no distinction between serum levels of D-dimer in patients who needed or did not need bowel resection due to AMI [49].

Icoz et al. tested the hypothesis that D-dimer serum levels function as a reliable marker for accurately diagnosing strangulated intestinal hernia [50]. The serum D-dimer levels were significantly higher in patients with mesenteric ischemia than in patients belonging to the control group [50]. Additionally, these levels correlated well with white blood cell count and LDH levels in patients with mesenteric ischemia [50]. The study also concluded that people with incarcerated or strangulated hernia needing bowel resection had insignificantly higher levels of D-dimer (*p* > 0.05) [50].

Another study conducted by Block et al. showed that no patient without mesenteric ischemia exhibited elevated D-dimer levels [51]. Moreover, other markers, such as alkaline liver phosphatase and certain isoenzymes of lactate dehydrogenase, also showed relatively high specificity and selectivity [51]. Additionally, D-dimer levels were positively associated with intestinal ischemia. As no patient diagnosed with a normal D-dimer presented with intestinal ischemia, the authors concluded that D-dimers may be used as an exclusion test for intestinal ischemia [51]. Table 1 presents the detailed results of this study.

Destek et al. conducted a study on 44 patients diagnosed with different etiologies of AMI, and several markers were monitored and analyzed [19]. The study concluded that, for AMI of embolic etiology and NOMI, the diagnosis could be sustained by a value of D-dimer of over 1.73 µg/mL FEU (fibrinogen equivalent units) or CRP > 9 mg/L [19]. Regarding AMI involving the thrombosis or embolism of the mesenteric artery, the neutrophil–lymphocyte ratio and CRP can be used, with values over 12.5 × 10^3^/µL and 19.4 mg/L, respectively, indicating a positive diagnosis [19].

One of the most important factors in the management of AMI is represented by early diagnosis. Thus, combining monitoring biomarkers and various imaging technologies has led to a more accurate and rapid diagnosis. Akyildiz et al. combined the level of D-dimer and biphasic CT with mesenteric CT angiography in 47 patients, out of which 28 were diagnosed with AMI [52]. The study concluded that D-dimer levels greater than 3.17 µg FEU/mL acted similarly to biphasic CT with mesenteric CT angiography when diagnosing AMI [52]. Gün et al. also compared the efficiency of D-dimer testing in patients presenting with suspicion of AMI to angio-CT [53]. Out of 676 patients who presented to the hospital with abdominal pain of various etiologies, 29 had positive abdominal findings on the CT scan. In comparison, 124 people had elevated D-dimer levels [53]. In addition, 13 patients were diagnosed with AMI with angio-CT, and 11 had elevated D-dimer levels [53].


**Alpha Glutathione S-transferase**


The glutathione S-transferase (GST) group is a family of multifunctional intracellular proteins involved in cell protection, detoxification, and antioxidation. There are four subtypes that display tissue variation, with alpha-GST being found in liver and intestinal cells. Upon intestinal cellular damage, alpha-GST is released into the circulation [54]. In a study including 26 patients with acute abdominal pain, 12 of whom had AMI, plasma alpha-GST was found to be significantly increased in patients with AMI [55]. Moreover, in a prospective study conducted by Elshoura et al., 90 patients were included after clinical suspicion of AMI, confirmed by CT angiography or laparotomy [56]. Patients with intestinal ischemia (n = 52) had significantly higher serum values of alpha-GST, with this marker diagnosing AMI with an accuracy of 84.4% [56]. Additionally, Block et al. investigated a series of biomarkers, including alpha-GST, D-dimer, alkaline phosphatase, and lactate dehydrogenase. Although specificity was high, sensitivity was relatively low in comparison to the other biomarkers (Table 1) [51].

### 3.2. Biomarkers in Preclinical Research


**Omentin**


Sit et al. used an animal model to observe whether serum omentin in rats could be an early predictor of AMI [48]. Rats were equally divided into three groups (sham, transient, and permanent ischemia) and underwent laparotomy and exposure, transient occlusion (45 min), or ligature of SMA. The results showed a significant decrease in serum omentin levels in rats subjected to temporary occlusion of the SMA compared to the sham group (*p* = 0.004). However, these findings were overshadowed by a lack of statistical significance regarding serum omentin differences between the permanent ischemia and sham groups. Therefore, it is difficult to appreciate the clinical significance of serum omentin, which acts as a biomarker for early-onset intestinal ischemia, as data from human patients are currently insufficient [48].

## 4. Discussion

Acute mesenteric ischemia (AMI) is one of the most severe causes of acute surgical abdomen. It is a condition that is equally difficult to diagnose as it is difficult to treat, with an often-unfavorable evolution and prognosis. Since it is a rare cause of abdominal pain, it often remains untreated, and in these conditions, the mortality rate reaches 50% [57].

A delayed diagnosis may lead to intestinal necrosis, septic shock, and multiple organ failure, which is life threatening in acute intestinal ischemia patients. Clinically, an accurate diagnosis of acute intestinal ischemia is very difficult even for a highly experienced physician. It is often diagnosed only after being found intraoperatively [58].

Establishing a set of biomarkers with a predictive and/or diagnostic role would be extremely useful in cases of AMI. Serious steps are being taken in this regard. At the present time, numerous studies have been published, but a clear conclusion has not yet been reached. It is certain that parameters such as citrulline, intestinal fatty acid-binding protein (I-FABP), D-lactate, along with D-dimers can be classified as sensitive rather than specific biomarkers in the diagnosis of AMI.

A comprehensive review of literature revealed a multitude of serum biomarkers explored in the context of mesenteric ischemia. The diverse nature of these biomarkers underscores the complexity of mesenteric ischemia, involving not only vascular compromise (endothelin-1, FAR, D-dimers, and ischemia-modified albumin) but also inflammatory (CRP, IL-6, NLR, PLR, procalcitonin, and omentin), and tissue damage processes (I-FABP, lactate, citrulline, and alpha-GST).

Several studies demonstrated the diagnostic utility of serum biomarkers in distinguishing between different etiologies of mesenteric ischemia, aiding clinicians in prompt and accurate diagnoses. Additionally, the prognostic value of these biomarkers in predicting the clinical outcomes of patients during follow-up emerged as a prominent theme. Elevated levels of certain biomarkers, such as D-dimer, were consistently associated with increased mortality and adverse outcomes.

Molecules that are related to vascular changes, such as endothelin-1, are valuable as biomarkers of bowel ischemia, but also as potential therapeutic targets, because of the damaging effects they produce, such as vasoconstriction, fibrosis of vascular cells, and the production of oxygen species, facilitating a significant decrease in mucosal blood flow and bowel necrosis [59]. Additionally, D-dimers can also have multiple utilities, such as a diagnostic role, high levels suggesting an ongoing clot formation process. However, D-dimers are not specific, as they can be increased in other conditions, such as deep vein thrombosis and pulmonary embolism. Thus, D-dimers can be used as a diagnostic marker together with other clinical biomarkers and imaging findings [57]. Another emerging biomarker is ischemia-modified albumin, a modified version of albumin, generated under ischemic conditions. Initially, the cobalt-binding capability of albumin was proposed for the assessment of IMA and demonstrated a high utility, mainly in myocardial ischemia. This marker can also be used in other conditions associated with ischemia and oxidative stress. Thus, ischemia-modified albumin serves as a sensitive but non-specific marker of the ischemic condition, including IMA [57].

Inflammatory biomarkers play an important role in diagnosis and prognosis in IMA. Although CRP and IL-6 play an important role in the inflammatory response during bowel ischemia, their usefulness as biomarkers is limited. Additionally, markers such as neutrophil–lymphocyte and platelet–lymphocyte ratios are informative of inflammatory and ischemic processes that might take place and are sensitive but are not specific to IMA. Ideally, they should be used in a multimodal approach, including other biomarkers, and imaging and clinical features.

Other markers such as procalcitonin show promise as a potential marker for intestinal ischemia, while the role of omentin in bowel ischemia is less well defined and has mainly been investigated in preclinical studies [57].

The most specific, however, are biomarkers derived from the components of enterocytes, released into circulation upon necrosis and ischemia. I-FABP is a useful tool in IMA; however, its diagnostic accuracy varies based on the extension of the ischemia and the time of the measurement after ischemia onset. Additionally, a single measurement may not be enough to diagnose or exclude IMA. Also, I-FABP can be used as a prognostic marker, with higher levels suggesting worse outcomes [60]. Another useful marker of hypoxia and ischemia is lactate. High levels of lactate can indicate ischemia even before tissue necrosis occurs. Thus, serum lactate is frequently used in clinical settings to diagnose intestinal ischemia; however, its sensitivity may not be high, as lactate levels do not increase very early during ischemia and its specificity is questionable, as lactate can increase in other types of conditions (sepsis, shock, etc.) [57]. Another interesting marker for bowel integrity is citrulline, an amino acid produced primarily by the small intestine enterocyte. Citrulline has been investigated for both diagnostic and prognostic utility, with higher plasma levels indicating a higher 28-day mortality, especially in combination with high levels of CRP. Also, monitoring citrulline kinetics over time could be useful for the prognosis of IMA patients [61]. In addition to citrulline, alpha-GST is another important constituent of enterocytes. Physiologically, it is involved in cell detoxification, but high plasma levels of alpha-GST indicate intestinal injury and ischemia [56]. While single biomarkers have clinical utility, there are currently studies investigating the power of biomarker panels, where several biomarkers are being determined at the same time. This approach can increase the strength of the diagnostic utility of each biomarker [57].

The strength of this research resides in its status as the inaugural systematic review analyzing the dynamics of biomarkers used for the diagnosis of mesenteric ischemia. Moreover, it extends its purview to assess the efficacy of serum biomarkers in prognosticating clinical outcomes during subsequent follow-up. Table 2 offers a more detailed view of the methodology and clinical utility of the results of the studies included in the present review.

Despite the promising findings, it is crucial to acknowledge the limitations inherent in the current body of evidence. Heterogeneity among studies in terms of patient populations, study designs, and methodologies may contribute to variations in the reported outcomes. Moreover, the lack of standardized cutoff values for biomarkers and the variability in the timing of measurements during follow-up pose challenges to their clinical applicability.

The integration of serum biomarkers into routine clinical practice for mesenteric ischemia necessitates further research to establish standardized protocols and validate their clinical significance. The potential benefits of early risk stratification, timely intervention, and improved patient outcomes underscore the importance of ongoing investigations in this field. Clinicians must carefully consider the context of each patient and interpret biomarker results in conjunction with clinical and imaging findings.

Future research efforts should focus on the development of a consensus regarding the optimal panel of biomarkers for mesenteric ischemia. Prospective studies with larger sample sizes and standardized methodologies are warranted to validate the findings of this systematic review. Additionally, exploring the potential of novel biomarkers and emerging technologies, such as omics approaches, may provide further insights into the pathophysiology and prognosis of mesenteric ischemia. Continued research endeavors are essential to refine the clinical utility of serum biomarkers and enhance their integration into routine practice for the benefit of patients with mesenteric ischemia.

However, a subset of serum markers that exhibit high levels of potential in accurately correlating with AMI still need to be thoroughly studied through rigorous human trials to have the capacity to fully predict the clinical results of patients with AMI during follow-up.

## 5. Conclusions

Specific biomarkers offer potential diagnostic and prognostic value, but the heterogeneity among studies underscores the necessity for standardized methodologies. The most studied diagnosis and/or prognosis biomarkers for AMI are those involved in vascular compromise (endothelin-1, FAR, D-dimers, and ischemia-modified albumin), inflammatory response (CRP, IL-6, NLR, PLR, procalcitonin, and omentin), and tissue damage (I-FABP, lactate, citrulline, and alpha-GST). These biomarkers emerged in a very intuitive way, considering the most important pathological processes that take place during AMI. However, further studies, even those without a physio-pathological hypothesis could result in the discovery of surprising new biomarkers.

The integration of validated biomarkers could enhance risk stratification and guide personalized therapeutic interventions, improving the overall management of mesenteric ischemia patients.

## Figures and Tables

**Figure 1 diagnostics-14-00670-f001:**
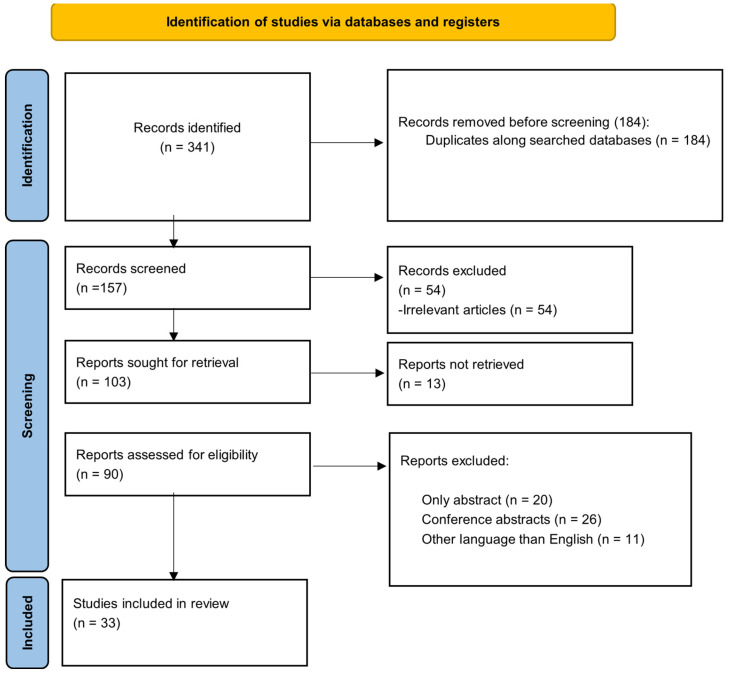
PRISMA flowchart of the included studies.

**Table 1 diagnostics-14-00670-t001:** Summary of results for biomarkers with clinical use.

Marker	Authors (Year)	Number of Patients	AU-ROC	Threshold (Unit)	Sensitivity (%)	Specificity (%)	PPV (%)	NPV (%)
Endothelin-1	Groesdonk et al. (2015)	865	0.77	14.5 pg/mL	51	94	NA	NA
I-FABP	Bourcier et al. (2022)	61	0.83	3114 pg/mL	70	85	90	58
	Sekino et al. (2017)	57	0.73	19 ng/mL	61.5	86.4	57.1	88.4
	Matsumoto et al. (2019)	96	0.805	15.5 ng/mL	76	80.3	57.6	90.5
	Güzel et al. (2014)	77	NA	90 pg/mL	90	100	100	87
	Kanda et al. (2011)	242	0.792	3.1 ng/ml	78.8	73.8	33.6	95.4
Serum lactate	Conde Monroy et al. (2022)	74	0.805	3.8 ng/ml	81	76	NA	NA
PCT	Nagata et al. (2008)	93	NR	2 ng/mL	100	83.9	27	100
	Markogiannakis et al. (2011)	242	0.77-ischemia 0.87-necrosis	0.25 ng/mL	72-ischemia 83-necrosis	73-ischemia 78-necrosis	60-ischemia 52-necrosis	83-ischemia 95-necrosis
	Cosse et al. (2013)	166	0.91	0.57 ng/mL	83.3	91.3	83.3	91.3
	Cosse et al. (2014)	59	0.86	0.53 ng/mL	80	84.8	40	90.7
Citrulline	Kulu et al. (2017)	48	0.72	15.8 µmol/L	39	100	100	64
	Nuzzo et al. (2021)	129	0.67	16.6 µmol/L	56	84	68	75
IL-6	Sgourakis et al. (2013)	56	1	27.66 pg/mL	100	100	NR	NR
	Salim et al. (2017)	20	0.85	0.04 ng/mL	100	60	NR	NR
IMA	Polk et al. (2008)	26	0.946	0.35 ABSU	100	86	100	85.7
	Roy et al. (2004)	131	0.78	93.5 U/mL	75	74.6	NR	NR
D-dimer	Chiu et al. (2009)	67	0.64	1.0 μg FEU/mL	96	18	NR	NR
	Icoz et al. (2006)	159	NR	NR	85	41	NR	94
	Block et al. (2008)	71	NA	0.9 mg/L	60	82	33	50
	Akyildiz et al. (2009)	47	0.93	3.17 μg FEU/mL	94.7	78.6	75	95.7
	Gün et al. (2014)	676	NR	1000 ng/mL	84.6	47.9	NR	NR
	Acosta et al. (2004)	101	NR	0.3 mg/L	100	36	13	100
α-GST	Block et al. (2008)	71	NA	4 ng/mL	20	87	15	90
	Delaney et al. (1999)	26	NR	4 ng/mL	100	86	86	100
	Elshoura et al. (2018)	90	NR	4 ng/mL	88.4	78.9	85.1	83.3

PPV—positive predictive value; NPV—negative predictive value; I-FABP—intestinal fatty acid-binding proteins; PCT—procalcitonin; IMA—ischemia-modified albumin; α-GST—alpha isoenzyme of glutathione S-transferase; FEU—fibrinogen equivalent units.

**Table 2 diagnostics-14-00670-t002:** Comparison of results.

Author (Year)	Study Design and Number of Patients	Analyzed Biomarkers	Conclusions
Martin (2023) [18]	Retrospective study (n = 119)	Serum lactate	After multivariate analysis, there was no statistically significance of plasma lactate association with delayed diagnosis.
Studer (2015) [17]	Retrospective study (n = 91)	Serum lactate	The highest serum lactate value measured within 24 h before surgery showed a moderate correlation, but no statistical significance with the length of bowel necrosis.
Conde Monroy (2022) [16]	Retrospective cross-sectional study with a prospective database (n = 74)	Serum lactate	Serum lactate admission levels can be considered as a useful prognostic tool in term of mortality. No statistically significant correlation between SLAL and bowel necrosis length.
Sekino (2017) [12]	Prospective observational study (n = 57)	I-FABP	I-FABP levels at ICU admission can serve as a predictor of 28-day mortality in septic shock patients and are associated with the incidence of NOMI.
Bourcier (2022) [11]	Prospective observational study (61 patients)	I-FABP Citrulline	Elevated plasma I-FABP is associated with the diagnosis of intestinal necrosis. Increased survival when necrosis resection was performed. Citrulline levels were not statistically significant in predicting necrosis and outcome.
Muhtaroğlu (2023) [28]	Retrospective study (n = 91)	FAR HGB CRP D-dimer WBC Neutrophils	The FAR ratio may be a valuable prognostic biomarker for patients with AMI. Pre- and postoperative levels of the following: Fibrinogen—statistically significantly higher in non-survivors (*p* < 0.001); Albumin—statistically significantly lower in non-survivors than in the survivors (*p* < 0.059, *p* < 0.001); FAR ratios—considerably higher in non-survivors (*p* < 0.001); Pre-HGB—statistically significantly higher in the survivor group. Postop-CRP, pre-WBC, pre-Neutrophil, and postop-D-dimer were statistically significantly higher in the non-survivors.
Destek (2020) [19]	Retrospective study (n = 44)	CRP L-lactate D-dimer WBC NLR	CRP level can be used effectively in the preoperative period to diagnose all etiological types of AMI. L-lactate, D-dimer, leukocyte, and NLR have no predictive value in the diagnosis of all AMI subtypes.
Augène (2019) [29]	Retrospective study (n = 106)	NLR PLR	The PLR value at the in-hospital admission is a reliable and simple predictive factor of 30-day mortality in patients with AMI. NLR showed not to be statistically significant.
Gunduz et al. (2008) [43]	Preliminary case control study	IMA	Statistically significant increases in IMA were observed in the occlusion group (n = 7) when compared to the control group (n = 7).
Cosse (2015) [30]	Retrospective, multicenter study (n = 128)	Procalcitonin (PCT)	PCT could be used as a marker of necrosis; especially in case of extended damages and reflects the patient’s prognosis.

PLR—platelet–lymphocyte ratio; NLR—neutrophil–lymphocyte ratio; I-FABP—intestinal fatty acid-binding proteins; SLAL—serum lactate admission levels; NOMI—non-occlusive mesenteric ischemia; FAR—fibrinogen–albumin ratio; AMI—acute mesenteric ischemia; IMA—ischemia-modified albumin; CRP—C reactive protein; WBC—white blood cells; HGB—hemoglobin.

## Data Availability

The original contributions presented in the study are included in the article, further inquiries can be directed to the corresponding author.

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
