# Peer review of "The Efficiency of Serum Biomarkers in Predicting the Clinical Outcome of Patients with Mesenteric Ischemia during Follow-Up: A Systematic Review"

_diagnostics, 2024, doi:10.3390/diagnostics14070670_

Round 1
Reviewer 1 Report
Comments and Suggestions for Authors
Dear authors,
The chosen topic is very useful, considering the clinician point of view.
However, some aspects should be addressed.
1. Please carefully see the PRISMA 2020 item checklist
2. Please state more explicit the rationale of the present review
3.Also provide a separate paragraph with the objectives
4.Please specify the methods used in order to assess the risk of bias in the included studies.
5. Please provide separate paragraphs for the inclusion and exclusion criteria.
6. Fig 1 related to the study selection is part of the Results Chapter and should be moved accordingly
7. Please provide more info related to the assessment the risk of bias of missing results , as well as certainty of evidence.
8. A quality assessment subchapter might be useful.
9. Except the two tables, the Results chapter with the analysis of the several biomarkers lacks the statistical details.
10. Conclusions should state more explicit which biomarkers seemed to be more accurate in the clinical outcome setting, according to the results of this study.
Author Response
Please see the attachment below.

Reviewer 2 Report
Comments and Suggestions for Authors
I read with interest a very long and draggy review article discussing role of various serological markers portrayed as molecular fingerprints to aid in diagnosis, predict prognosis and possibly guide treatment in patients with acute mesenteric ischaemia.
While in general, this is well written and a clear manuscript, i find three main issues which authors should consider to address:
1. Manuscript is too long with lot of details of each marker. I recommend to trim the content, reduce the citation count and group the markers as 'clinical use' and 'research limited' markers as some markers are not universally available. Authors can use their institute as a benchmark for this division and mention this in method section. Than focus more on clinical and less on research as your primary aim is clinical diagnosis, prognosis etc.
2. I find that authors have blurred the boundary between a 'narrative' review versus 'systematic' review. This manuscript is actually more of a narrative review and less of a systematic review of 33 studies that PRISMA generated. So either omit the PRISMA and make this a narrative review (which it is) or trim the content substantially (> one third of details and many citations) and focus on the 33 studies to make it a systematic review.
3. See if some of the text can be converted to a figure or a graphic and thus improve the layout when the paper is eventually published. An average reader is unlikely to read long worded paragraphs and even if you will manage to publish, unlikely it will serve readers or your motive to spread awareness.
Author Response
Please see the attachment below.

Round 2
Reviewer 2 Report
Comments and Suggestions for Authors
nil